# Navigating Semantic Drift in Task-Agnostic Class-Incremental Learning

**Fangwen Wu** [1]   **Lechao Cheng**[✉ 2]   **Shengeng Tang** [2]   **Xiaofeng Zhu** [1]   **Chaowei Fang** [3]
**Dingwen Zhang**[✉ 4]   **Meng Wang** [2]

## Abstract

Class-incremental learning (CIL) seeks to enable a model to sequentially learn new classes while retaining knowledge of previously learned ones. Balancing flexibility and stability remains a significant challenge, particularly when the task ID is unknown. To address this, our study reveals that the gap in feature distribution between novel and existing tasks is primarily driven by differences in mean and covariance moments. Building on this insight, we propose a novel semantic drift calibration method that incorporates mean shift compensation and covariance calibration. Specifically, we calculate each class's mean by averaging its sample embeddings and estimate task shifts using weighted embedding changes based on their proximity to the previous mean, effectively capturing mean shifts for all learned classes with each new task. We also apply Mahalanobis distance constraint for covariance calibration, aligning class-specific embedding covariances between old and current networks to mitigate the covariance shift. Additionally, we integrate a feature-level self-distillation approach to enhance generalization. Comprehensive experiments on commonly used datasets demonstrate the effectiveness of our approach. The source code is available at https://github.com/fwu11/MACIL.git.

## 1. Introduction

Continual Learning, also referred to as lifelong learning, aims to enable machine learning models to sequentially learn multiple tasks over their life-cycle without requiring retraining or access to data from previous tasks (Rebuffi

[1]Zhejiang Lab [2]Hefei University of Technology [3]Xidian University [4]Northwestern Polytechnical University. Correspondence to: Lechao Cheng <chenglc@hfut.edu.cn>, Dingwen Zhang <zhangdingwen2006yyy@gmail.com>.

*Proceedings of the 42nd International Conference on Machine Learning*, Vancouver, Canada. PMLR 267, 2025. Copyright 2025 by the author(s).

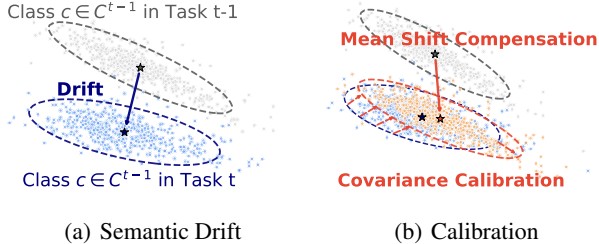

(a) Semantic Drift        (b) Calibration

*Figure 1.* As new tasks are learned, the categories from previous tasks in the latest updated model continuously experience shifts in their means and variances, referred to as (a) **Semantic Drift**. In this paper, we calibrate such semantic drift by applying explicit mean shift compensation and implicit variance constraints (b).

et al., 2017). The primary goal of continual learning (Cheng et al., 2024a;b; Zhang et al., 2024; Huang et al., 2024a;b) is to facilitate knowledge accumulation and transfer, allowing models to adapt quickly to new, unseen tasks while maintaining robust performance on previously learned tasks (Parisi et al., 2019). This capability has broad applications in fields such as computer vision, robotics, and natural language processing.

In recent years, the issue of model plasticity has become less prominent in deep learning-based approaches, primarily due to two factors: (1) the increasing capacity of deep models allows them to effectively over-fit new data, and (2) large-scale pre-training on extensive datasets equips models with powerful feature extraction capabilities (He et al., 2019). Parameter-efficient fine-tuning based on pretrained models has further enhanced model plasticity, as highlighted in numerous recent studies (Houlsby et al., 2019; Lester et al., 2021; Hu et al., 2022).

Despite these advancements, existing methods—such as regularization (Kirkpatrick et al., 2017), memory replay (Lopez-Paz & Ranzato, 2017), and knowledge distillation (Li & Hoiem, 2017)—while improving stability to some extent, introduce additional costs. For example, (1) memory replay methods require storing and retraining on old task data, increasing storage and computational demands. (2) knowledge distillation involves additional computational overhead during the distillation process, complicating and slowing down training. These additional costs hinder the

practical deployment of continual learning methods. Therefore, a key challenge in the field is to improve model stability while minimizing resource consumption and computational overhead (Wang et al., 2024b).

In this paper, we build upon successful practices by leveraging parameter-efficient fine-tuning based on pretrained models to further analyze catastrophic forgetting. Through extensive experimental observations, we discovered that although low-rank adaptation (e.g., LoRA (Hu et al., 2022) ) based on pretrained models can effectively maintain model plasticity, the incremental integration of tasks and model updates induces feature mean and covariance shift—what we also term **Semantic Drift**. As illustrated in Figure 1, the feature distribution of the original task data undergoes significant mean shifts and changes in variance shape as more tasks are introduced.

Based on these observations, we propose to address semantic drift with both mean shift compensation and covariance calibration, which constrain the first-order and second-order moments of the features, respectively. Specifically, we compute mean class representations after learning a novel task as the average embedding of all samples in class. The shift between old and novel tasks is approximated by the weighted average of embedding shifts, where the weights are determined by the proximity of each embedding to the previous class mean. This approach effectively estimates the mean shift for all previously learned classes during each new task. We also introduce an implicit covariance calibration technique using Mahalanobis distance (Mahalanobis, 1936) loss to address semantic drift. This method aligns the covariance matrices of embeddings from old and current networks for each class, ensuring consistent intra-class distributions. By leveraging the old network as "past knowledge", we compute class-specific covariance matrices and minimize the absolute differences in Mahalanobis distances between embedding pairs from both networks. This approach effectively mitigates covariance shift, maintaining model stability while allowing continual learning. As shown in Figure 1(b), these constraints effectively maintain model stability while preserving plasticity. Additionally, we implement feature self-distillation for patch tokens, further enhancing feature stability. In summary, our main contributions are:

- We delve into the exploration of the semantic drift problem in class-incremental learning and further propose efficient and straightforward solutions—mean shift compensation and covariance calibration, which significantly alleviate this challenge.

- We orchestrate an efficient task-agnostic continual learning framework that outperforms existing methods across multiple public datasets, demonstrating the superiority of our approach.

## 2. Related Works

### 2.1. Class-Incremental Learning

Class-Incremental Learning (CIL) aims to enable a model to sequentially learn new classes without forgetting previously learned ones. This presents a significant challenge, especially since task-IDs are not available during inference. To address this issue, several strategies have been proposed, which can be broadly categorized as follows:

Regularization-based methods (Li & Hoiem, 2017; Rebuffi et al., 2017; Kirkpatrick et al., 2017; Zenke et al., 2017) focus on constraining changes to important model parameters during training on new classes. Replay-based methods address forgetting by maintaining a memory buffer that stores examples from prior tasks. When learning a new task, the model is trained not only on the current task but also on these stored examples, helping it retain previous knowledge. These methods include direct replay (Lopez-Paz & Ranzato, 2017; Riemer et al., 2019; Chaudhry et al., 2019; Liu et al., 2021) as well as generative replay (Shin et al., 2017; Zhu et al., 2021). Optimization-based methods focus on explicitly designing and modifying the optimization process to reduce catastrophic forgetting, such as through gradient projection (Farajtabar et al., 2020; Saha et al., 2021; Lu et al., 2024) or loss function adjustments (Wang et al., 2021; Wen et al., 2023). Representation-based methods aim to maintain a stable and generalizable feature space as new classes are added. These include self-supervised learning (Cha et al., 2021; Pham et al., 2021) and the use of pre-trained models (Wang et al., 2022a; Gao et al., 2024a; McDonnell et al., 2023). A key challenge for exemplar-free methods is the shift in backbone features. Recent studies have proposed estimating this shift through changes in class prototypes (Yu et al., 2020; Gomez-Villa et al., 2024; Goswami et al., 2024). This work investigates the semantic drift phenomenon in both the mean and covariance, calibrating them to mitigate catastrophic forgetting.

### 2.2. Pre-trained Model based Class-Incremental Learning

Pre-trained models have become a key component in CIL due to their ability to transfer knowledge efficiently. It is prevailing to use parameter-Efficient Fine-Tuning (PEFT) methods to adapt the model computation efficiently. PEFT methods introduce a small portion of the learnable parameters while keeping the pre-trained model frozen. LoRA (Hu et al., 2022) optimizes the weight space using low-rank matrix factorization, avoiding full parameter fine-tuning; VPT (Jia et al., 2022; Wang et al., 2024d) injects learnable prompts into the input or intermediate layers to extract task-specific features while freezing the backbone network; AdaptFormer (Chen et al., 2022), based on adaptive Transformer components, integrates task-specific informa-

tion with general knowledge.

Prompt-based class-incremental continual learning methods dynamically adjust lightweight prompt parameters to adapt to task evolution. Key mechanisms include: dynamic prompt pool retrieval (Wang et al., 2022c), general and expert prompt design for knowledge sharing (Wang et al., 2022b), discrete prompt optimization (Jiao et al., 2024), consistency alignment between classifiers and prompts (Gao et al., 2024b), decomposed attention (Smith et al., 2023), one forward stage (Kim et al., 2024), and evolving prompt adapting to task changes (Kurniawan et al., 2024). Adapter-based methods such as EASE (Zhou et al., 2024b) dynamically expand task-specific subspaces and integrate multiple adapter predictions with semantic-guided prototype synthesis to mitigate feature degradation of old classes; SSIAT (Tan et al., 2024) continuously tunes shared adapters and estimates mean shifts, updating prototypes to align new and old task features; and MOS (Sun et al., 2025) merges adapter parameters and employs a self-optimization retrieval mechanism to optimize module compatibility and inference efficiency. LoRA-based methods, such as InfLoRA (Liang & Li, 2024), introduce orthogonal constraints to isolate low-rank subspaces, effectively reducing parameter interference between tasks. Together, these methods offer efficient and scalable solutions for adapting pre-trained models to class-incremental learning tasks. In this work, we leverage the power of the LoRA in the context of CIL and build our semantic drift calibration modules on top of it.

## 3. Method

### 3.1. Preliminaries

#### 3.1.1. CLASS-INCREMENTAL LEARNING

Consider a dataset consisting of $T$ tasks $\{\mathcal{D}^t\}_{t=1}^T$. For each task, the dataset $\mathcal{D}^t = \{(x_i^t, y_i^t)\}_{i=1}^{n^t}$ contains $n^t$ inputs $x_i^t \in \mathbb{R}^d$ and their corresponding labels $y_i^t \in C^t$. We use $X^t$ and $Y^t$ to denote the collection of input data and label of task $t$, respectively and $C^t = \{c_i\}_{i=1}^{|C^t|}$ is the label set contains $|C^t|$ classes. In the class-incremental setting, for any task $i \neq j$, the input data from different tasks follow different distributions, i.e., $p(X^i) \neq p(X^j)$ and labels satisfy $C^i \cap C^j = \emptyset$. The learning objective is to find the model $f^t : \mathbb{R}^D \to \mathbb{R}^{C^t}$, where $\mathcal{C}^t = \cup_{i=1}^t C^i$ represents the total number of classes learned. This model is trained on all training datasets to perform well on all test dataset seen up to task $t$. In our scenario, the model $f^t$ is based on a pre-trained model, consisting of $f_\theta^t(x) = W^\top \phi_\theta^t(x)$, where $\phi_\theta^t : \mathbb{R}^D \to \mathbb{R}^d$ is a feature extractor composed of a a frozen pre-trained model $\phi$ and learnable parameters $\theta$ in the LoRA modules, and a classification head $W = \{w^t\}_{t=1}^T$ for each task, where we have $w^t \in \mathbb{R}^{d \times C_t}$. For a given task $t$, the old network $f_\theta^{t-1}(x)$ refers to the network trained on

task $t-1$, and it is frozen in task $t$.

### 3.2. Overview

Figure 2 illustrates the overall architecture for class-incremental learning. We employ a frozen pre-trained ViT (Dosovitskiy et al., 2021) model as the backbone with learnable task-specific LoRA modules. The output class tokens are forwarded through a task-specific classifier, which generates the class scores, while the angular penalty loss (Peng et al., 2022; Tan et al., 2024) is used to compute the classification loss $\mathcal{L}_{\text{cls}}$:

$$\mathcal{L}_{\text{cls}} = -\frac{1}{n^t} \sum_{j=1}^{n^t} \log \frac{\exp(s\cos(\theta_j))}{\sum_{i=1}^{|C^t|} \exp(s\cos(\theta_i))} \qquad (1)$$

where $\cos(\theta_j) = \frac{w_j f_{\theta_j}}{\|w_j\|\|f_{\theta_j}\|}$, $s$ represents the scaling factor, and $n^t$ is the number of training samples in task $t$. The mean and covariance of each class are stored for each learning session. Before the training process, covariance of each class is precomputed from the class tokens generated by the network trained on previous tasks. These covariance matrices are then used to align the distribution of the representations generated by the current network with that of the old network, based on the Mahalanobis distance. This is referred to as the covariance calibration loss $\mathcal{L}_{\text{cov}}$. Furthermore, patch tokens are leveraged to preserve knowledge from earlier tasks at the feature level through a distillation loss $\mathcal{L}_{\text{distill}}$.

After the training process, the class means are updated through the mean shift compensation process, and the classifier heads are retrained using the calibrated class statistics. The training pipeline is illustrated in Algorithm 1. In summary, the overall training objective is:

$$\mathcal{L} = \mathcal{L}_{\text{cls}} + \mathcal{L}_{\text{cov}} + \lambda\mathcal{L}_{\text{distill}} \qquad (2)$$

### 3.3. Low-Rank Adaptation

In class-incremental learning (CIL), task IDs are not provided during the inference stage. For methods based on pre-trained models, the use of task-specific PEFT modules often involves a task ID prediction step during testing (Wang et al., 2022b; 2024a; Sun et al., 2025). Other approaches avoid task ID prediction, as low prediction accuracy can negatively impact performance. For instance, some methods use weighted sums of prompts to determine the prompts applied during inference (Kurniawan et al., 2024; Smith et al., 2023), or aggregate all previous PEFT modules (Zhou et al., 2024b; Liang & Li, 2024). Alternatively, some methods rely on a shared PEFT module across tasks (Huang et al., 2024c; Tan et al., 2024).

Compared to other PEFT modules applied in CIL, such as prompts and adapters, LoRA (Hu et al., 2022) performs

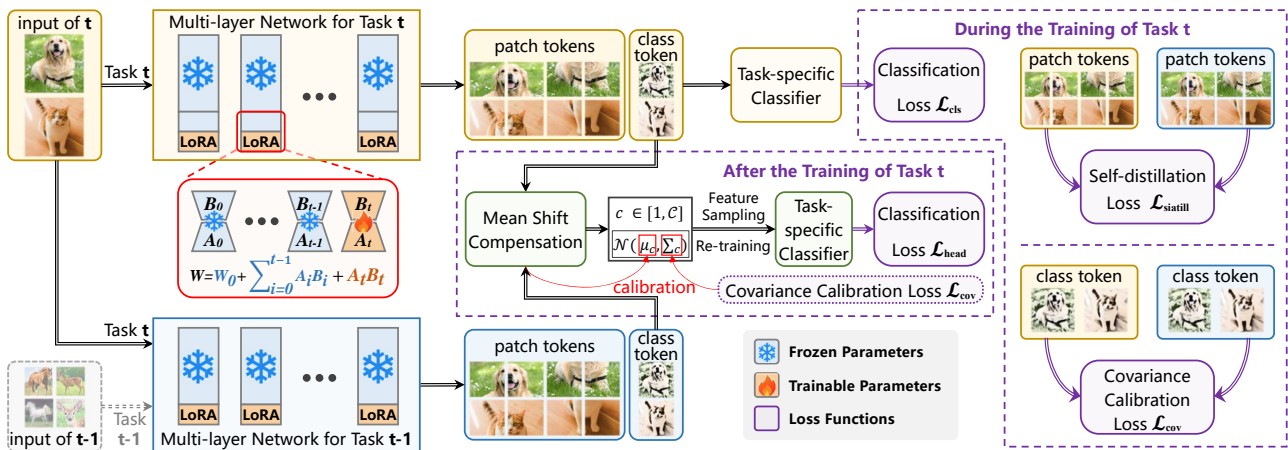

*Figure 2.* Illustration of our method at task $t$. The feature extractor at task $t$ uses a frozen pre-trained ViT backbone with learnable LoRA modules. The output class tokens (yellow) are passed through a classifier to compute the classification loss $\mathcal{L}_{\text{cls}}$, and the mean and covariance of each class are stored for each session. During training, class tokens (yellow and blue) are used to align class distributions via a covariance calibration loss $\mathcal{L}_{\text{cov}}$. Patch tokens from network $t$ (yellow) distill knowledge from network $t-1$ (blue) through a distillation loss $\mathcal{L}_{\text{distill}}$. After training, the class means are updated using a mean shift compensation module, and the classifier heads are retrained with the calibrated statistics.

inference by adding the low-rank adaptation matrices to the original weight matrices. This enables LoRA to efficiently combine the pre-trained model weights with the task-specific adaptation matrices, as well as the information across the different task-specific matrices.

In this paper, we utilize task-specific LoRA modules, where each task is assigned a unique LoRA module. Considering the pre-trained weight matrix $W_0 \in \mathbb{R}^{d \times k}$, the update of the weight matrix is decomposed into the product of the low rank matrices $B \in \mathbb{R}^{d \times r}$ and $A \in \mathbb{R}^{r \times k}$, where $r$ is a value much smaller than the input dimension $d$ and the output dimension $k$. The aggregation of all previous LoRA modules at task $t$ can be expressed as $W = W_0 + \sum_{i=1}^{t} B_i A_i$. Additionally, we explore two other LoRA structures: the task-shared LoRA module, which uses a common LoRA module for all tasks ($W = W_0 + BA$), and a hybrid structure that combines both task-specific and task-shared designs, inspired by HydraLoRA (Tian et al., 2024). In this hybrid structure, the shared LoRA module's matrix $A$ is used across tasks, while the independent LoRA module's matrix $B_i$ is specific to each task ($W = W_0 + \sum_{i=1}^{t} B_i A$). During training, only the low-rank weight matrices $A_i$ and $B_i$ are learnable, and the pre-trained weight $W_0$ is frozen.

### 3.4. Semantic Drift

As tasks increase, we no longer have access to the data from previous tasks, and therefore cannot compute the true distribution of earlier classes under the incrementally trained network. Both the mean and variance of feature distributions of the old classes change. This phenomenon is referred to as semantic drift. When semantic drift occurs, it affects the

classifier's performance. Thus, it is necessary to impose constraints on the semantic drift and calibrate the means and covariances of class distributions.

#### 3.4.1. MEAN SHIFT COMPENSATION

We define the mean class representation of class $c$ in the embedding space after learning task $t$ as:

$$\mu_c^t = \frac{1}{N_c} \sum_{i=1}^{N_c} [y_i = c] \phi_\theta^t(x_i) \tag{3}$$

where $N_c$ is the number of samples for class $c$. The shift between the class mean obtained with the current network and the class mean obtained with the previous network can be defined as:

$$\Delta \mu_c^{t-1 \to t} = \mu_c^t - \mu_c^{t-1} \tag{4}$$

Previous work (Yu et al., 2020) suggests that the shift between the true class mean and the estimated class mean can be approximated by the shift in the current class embeddings between the old and current models. Specifically, the shift of the class embeddings can be defined as:

$$\Delta \phi_\theta^{t-1 \to t}(x_i) = \phi_\theta^t(x_i) - \phi_\theta^{t-1}(x_i) \tag{5}$$

where $x_i$ belongs to class $c$. We can compute $\phi_\theta^{t-1}(x_i)$ using the model trained on task $t-1$ before the current task training. Then, we compute the drift $\phi_\theta^t(x_i)$ and use it to approximate the class mean shift $\Delta \mu_c^{t-1 \to t}$:

$$\hat{\Delta} \mu_c^{t-1 \to t} = \frac{\sum_i w_i \Delta \phi_\theta^{t-1 \to t}(x_i)}{\sum_i w_i} \tag{6}$$

with

$$w_i = \exp(-\frac{\|\phi_\theta^{t-1}(x_i) - \mu_c^{t-1}\|^2}{2\sigma^2}) \qquad (7)$$

where $\sigma$ is the standard deviation of the Gaussian kernel, and the weight $w_i$ indicates that embeddings closer to the class mean contribute more to the mean shift estimation of that particular class. The proposed method can be used to compensate the mean shift of all previously learned classes at each new task with $\mu_c^t \approx \hat{\mu}_c^t = \hat{\Delta}\mu_c^{t-1 \to t} + \mu_c^{t-1}$.

### 3.4.2. COVARIANCE CALIBRATION

In this section, we address semantic shift from the perspective of the covariance matrix by introducing a novel covariance calibration technique, which is powered by a Mahalanobis distance-based loss. The objective is to ensure that, for each class in the new dataset, the embeddings generated by both the old and current networks follow the same covariance structure. Specifically, the covariance matrices of the embeddings from the current network should be aligned with those from the old network. To achieve this, we utilize the old network, trained on the previous task, as a form of "past knowledge" and use it to calculate the covariance for each class in the current task. Since the Mahalanobis distance directly depends on the covariance matrix, optimizing the difference between embedding pairs from the old and current networks in terms of Mahalanobis distance implicitly constrains the shape of the intra-class distribution, thus alleviating covariance shift.

Mathematically, the Mahalanobis distance (Mahalanobis, 1936) is defined as the degree of difference between two random variables $x$ and $y$, which follow the same distribution and share the covariance matrix $\Sigma$:

$$d_M(x, y, \Sigma) = \sqrt{(x-y)^T \Sigma^{-1}(x-y)} \qquad (8)$$

In our setting, we calculate the Mahalanobis distance $d_M(\phi_\theta^t(x_i), \phi_\theta^t(x_j), \Sigma_c^{t-1})$ using the embedding pairs calculated with the data from the current task $t$ and as computed by covariance matrix $\Sigma_c^{t-1}$ of the class $c$ from the current task with the old network $f^{t-1}$.

Before training, for each class $c$, the covariance matrix is computed as:

$$\Sigma_c^{t-1} = \frac{1}{N_c} \sum_{i=1}^{N_c} (x_i - \mu_c^{t-1})(x_i - \mu_c^{t-1})^T \qquad (9)$$

where $\mu_c^{t-1}$ is the class mean of the class $c$ calculated with the old network $f^{t-1}$. The loss function for minimizing the absolute difference in Mahalanobis distances of the sample input pairs $(x_i, x_j)$ between embeddings obtained from the old and new networks:

$$\mathcal{L}_{\text{cov}} = \sum_{c \in C^t} \sum_{i,j} |d_M(\phi_\theta^t(x_i), \phi_\theta^t(x_j), \Sigma_c^{t-1}) \\ - d_M(\phi_\theta^{t-1}(x_i), \phi_\theta^{t-1}(x_j), \Sigma_c^{t-1})| \qquad (10)$$

### 3.5. Classifier Alignment

The model exhibits a tendency to prioritize the categories associated with the current task, resulting in a degradation of classification performance for categories from previous tasks. Upon completing the training for each task, the classifier undergoes post hoc retraining using the statistics of previously learned classes, thereby enhancing its overall performance. It is assumed that the feature representations learned by the pre-trained model for each class follow a Gaussian distribution (Zhang et al., 2023). In this framework, the mean $\mu_c$ and covariance $\Sigma_c$ of the feature representation for each class $c$, as described in previous sections, are calculated and stored. A number of $s_c$ samples $h^c$ are then drawn from the Gaussian distribution $\mathcal{N}(\mu_c, \Sigma_c)$ for each class as input. The classification head is subsequently retrained using a cross-entropy loss function:

$$\mathcal{L}_{head} = -\frac{1}{s_c|\mathcal{C}|} \sum_{i=1}^{s_c|\mathcal{C}|} \log\left(\frac{e^{w_j h_i^c}}{\sum_{k \in \mathcal{C}} e^{w_k(h_i^c)}}\right) \qquad (11)$$

where $\mathcal{C}$ denotes all classes learned until current task, $w$ is the classifier. With the alignment of semantic drift, the true class mean and covariance are calibrated, which helps mitigate the classifier's bias, typically induced by overconfidence in new tasks, and alleviates the issue of catastrophic forgetting.

### 3.6. Feature-level Self-Distillation

To enhance the model's resistance to catastrophic forgetting, we propose a self-distillation approach that focuses on improving the utilization of patch tokens in classification tasks, as suggested in (Li et al., 2024; Wang et al., 2024c). In Vision Transformers (ViT), the information from patch tokens is often underutilized, which can limit the model's ability to generalize across tasks (Zhai et al., 2024). To address this, we introduce a self-distillation loss based on patch tokens.

In this method, the class token output from the current network is treated as the essential feature information that needs to be learned for the current task. The feature-level self-distillation loss encourages the alignment of patch tokens from the current network output with the class token. Specifically, we compute the angular similarity, $\text{sim}_\angle$, between the current task's patch tokens, denoted as $p_j^t$, and the class token $c^t$, with the features normalized using L2

normalization. The loss is then formulated as:

$$\mathcal{L}_{\text{distill}} = \frac{1}{L} \sum_{j=1}^{L} \left(1 - \text{sim}_{\angle}(p_j^t, c^t)\right) \cdot \left\| p_j^t - p_j^{t-1} \right\|_2^2 \quad (12)$$

where $p_j^t$ is the $j$-th patch token for the current task, $c^t$ is the class token for the current task, $p_j^{t-1}$ is the patch token from the previous task, and $L$ is the total number of patch tokens for the current task. We disable the gradient updates during angular similarity computation.

The low angular similarity between patch tokens and class tokens suggests that the patch tokens contribute less to the semantic representation of the current task. To encourage better feature reuse, patch tokens with low similarity are encouraged to resemble the patch tokens from the previous network, thereby improving the retention of important task-related features. This approach ensures that patch tokens are more effectively utilized, contributing to the model's robustness and its ability to mitigate forgetting when transitioning between tasks.

---

**Algorithm 1** Semantic Drift Calibration

---

**Require:** Incrementally learned model $\{\phi_\theta^t\}_{t=1}^T$ with learning parameters $\theta$, classifiers $\{w^t\}_{t=1}^T$, dataset $\{D^t\}_{t=1}^T$;
1: **for** task $t = 1$ to $T$ **do**
2:     **for** $c \in C^t$ **do**
3:         Extract features $\phi_\theta^{t-1}(x_i)$ using the frozen model learned from task $t - 1$;
4:         Compute mean $\mu_c^{t-1}$ and covariance $\Sigma_c^{t-1}$;
5:     **end for**
6:     **for** Batch $\{(x_i^t, y_i^t)\}$ sampled from $\mathcal{D}^t$ **do**
7:         Train $\phi_\theta^t$ and $w^t$ using $\mathcal{L} = \mathcal{L}_{\text{cls}} + \mathcal{L}_{\text{cov}} + \lambda \mathcal{L}_{\text{distill}}$;
8:     **end for**
9:     **for** $c \in \cup_{i=1}^{t-1} C^i$ **do**
10:        Estimate and compensate the class mean shift $\mu_c^t \approx \hat{\mu}_c^t = \mu_c^{t-1} + \hat{\Delta}\mu_c^{t-1 \to t}$;
11:        Sample from $\mathcal{N}(\mu_c, \Sigma_c)$ with the calibrated statistics and retrain the classifiers $\{w^i\}_{i=1}^t$ with $\mathcal{L}_{\text{head}}$;
12:        Store mean $\mu_c$ and covariance $\Sigma_c$;
13:     **end for**
14: **end for**

---

# 4. Experiments

## 4.1. Setup

### 4.1.1. DATASETS AND METRICS.

We train and validate our method using four popular CIL datasets. ImageNet-R (Hendrycks et al., 2021a) is generated by applying artistic processing to 200 classes from ImageNet. The dataset consists of 200 categories, and we split ImageNet-R into 5, 10, and 20 tasks, with each task containing 40, 20, and 10 classes, respectively. CIFAR-100 (Krizhevsky, 2009) is a widely used dataset in CIL, containing 60,000 images across 100 categories. We also split CIFAR-100 into 5, 10, and 20 tasks with each task containing 20, 10, 5 classes, respectively. CUB-200 (Wah et al., 2011) is a fine-grained dataset containing approximately 11,788 images of 200 bird species with detailed class labels. ImageNet-A (Hendrycks et al., 2021b)is a real-world dataset consisting of 200 categories, notable for significant class imbalance, with some categories having very few training samples. We split CUB-200 and ImageNet-A into 10 tasks with 20 classes each.

We follow the commonly used evaluation metrics in CIL. We denote $a_{i,j}$ as the classification accuracy evaluated on the test set of the $j$-th task (where $j \leq i$) after learning $i$ tasks in incremental learning. The final accuracy is calculated as $\mathcal{A}_{\text{last}} = \frac{1}{t} \sum_{j=1}^{t} a_{i,j}$ and the average accuracy of all incremental tasks is $\mathcal{A}_{\text{avg}} = \frac{1}{T} \sum_{i=1}^{T} \mathcal{A}_i$. In line with other studies, our evaluation results are based on three trials with three different seeds. We report both the mean and standard deviation of the trials.

### 4.1.2. IMPLEMENTATION DETAILS.

In our experiment, we adopt ViT-B/16 (Dosovitskiy et al., 2021) pre-trained on ImageNet21K (Russakovsky et al., 2015) as the backbone. We use the SGD optimizer with the initial learning rate set as 0.01 and we use the Cosine Annealing scheduler. We train the first session for 20 epochs and 10 epochs for later sessions. The batch size is set to 48 for all the experiments. LoRA module is inserted to the key and value of all the attention layers in the transformer. The distillation loss weight $\lambda$ is set to 0.4, the LoRA rank $r$ is set to 32, and the scale $s$ in the angular penalty loss is set to 20. These values are determined through sensitivity analysis.

## 4.2. Comparison with State-of-the-arts

We conduct a comparative evaluation of our proposed method against state-of-the-art (SOTA) class-incremental learning (CIL) approaches based on pre-trained models, with a particular focus on techniques utilizing parameter-efficient fine-tuning (PEFT). To ensure a fair comparison, we evaluate all methods using the same ViT-B/16-IN21K pre-trained models, identical random seeds, and consistent class orders. Specifically, we compare prompt-based approaches, including L2P (Wang et al., 2022c), DualPrompt (Wang et al., 2022b), CODAPrompt (Smith et al., 2023), VQ-Prompt (Jiao et al., 2024), OS-Prompt (Kim et al., 2024), and CPrompt (Gao et al., 2024b); adapter-based methods such as SSIAT (Tan et al., 2024), EASE (Zhou et al., 2024b), and MOS (Sun et al., 2025); the LoRA-based method InfLoRA (Liang & Li, 2024) integrated with Classifier Alignment, referred to as InfLoRA+CA; the first-session adaptation method RanPAC (McDonnell et al., 2023), which

*Table 1.* Last and average performance results on four benchmark datasets (10 tasks) are reported. The mean and standard deviation of three trials are provided. We compare all methods using the same ViT-B/16-IN21K backbone, seeds, and class orders. For L2P and DualPrompt, we use the implementations provided by (Zhou et al., 2024a). Missing implementations on the datasets are denoted as '-'.

| Method | ImageNet-R | | ImageNet-A | | CUB-200 | | CIFAR-100 | |
|---|---|---|---|---|---|---|---|---|
| | $\mathcal{A}_{\text{Last}}$ | $\mathcal{A}_{\text{Avg}}$ | $\mathcal{A}_{\text{Last}}$ | $\mathcal{A}_{\text{Avg}}$ | $\mathcal{A}_{\text{Last}}$ | $\mathcal{A}_{\text{Avg}}$ | $\mathcal{A}_{\text{Last}}$ | $\mathcal{A}_{\text{Avg}}$ |
| L2P(Wang et al., 2022c) | 70.56±0.51 | 75.60±0.34 | - | - | - | - | 84.74±0.44 | 88.65±1.02 |
| DualPrompt(Wang et al., 2022b) | 66.89±0.70 | 71.60±0.31 | - | - | - | - | 85.17±0.55 | 89.18±1.01 |
| CODA-Prompt(Smith et al., 2023) | 72.82±0.50 | 78.13±0.52 | - | - | - | - | 87.00±0.31 | 90.68±1.02 |
| SLCA(Zhang et al., 2023) | 78.95±0.36 | 83.20±0.23 | - | - | 86.13±0.57 | 91.75±0.42 | 90.30±0.43 | 93.32±0.99 |
| RanPAC(McDonnell et al., 2023) | 77.94±0.14 | 82.98±0.31 | 62.25±0.35 | 69.92±1.69 | 89.94±0.52 | 93.68±0.48 | 92.09±0.13 | 94.75±0.64 |
| OS-Prompt(Kim et al., 2024) | 74.76±0.23 | 80.29±0.71 | - | - | - | - | 86.50±0.11 | 90.68±1.32 |
| VQ-Prompt(Jiao et al., 2024) | 75.68±0.23 | 80.02±0.18 | - | - | 86.47±0.40 | 91.37±0.54 | 90.27±0.06 | 93.10±0.84 |
| CPrompt(Gao et al., 2024b) | 76.38±0.46 | 81.52±0.38 | - | - | - | - | 87.63±0.17 | 91.50±1.10 |
| EASE(Zhou et al., 2024b) | 75.91±0.17 | 81.38±0.29 | 54.93±1.14 | 63.92±0.76 | 85.04±1.42 | 90.93±1.03 | 88.22±0.44 | 92.02±0.76 |
| InfLoRA+CA(Liang & Li, 2024) | 78.78±0.31 | 83.37±0.54 | - | - | - | - | 91.39±0.27 | 94.06±0.88 |
| SSIAT(Tan et al., 2024) | 79.55±0.27 | 83.70±0.38 | 62.65±1.28 | 71.14±1.24 | 89.68±0.48 | 93.67±0.46 | 91.41±0.14 | 94.27±0.75 |
| MOS(Sun et al., 2025) | 77.68±0.41 | 82.06±0.53 | 54.75±1.09 | 63.32±2.38 | 89.97±0.32 | 93.43±0.60 | 91.53±0.35 | 94.21±0.91 |
| **Ours** | 81.88±0.07 | 85.95±0.27 | 64.14±0.58 | 71.45±1.35 | 90.52±0.13 | 93.93±0.47 | 91.94±0.17 | 94.43±0.79 |

*Table 2.* Last and average results on ImageNet-R and CIFAR-100 are reported. The mean and standard deviation of three trials are provided for 5 and 20 tasks settings. We compare all methods using the same ViT-B/16-IN21K backbone, seeds, and class orders.

| Method | ImageNet-R | | | | CIFAR-100 | | | |
|---|---|---|---|---|---|---|---|---|
| | 5 tasks | | 20 tasks | | 5 tasks | | 20 tasks | |
| | $\mathcal{A}_{\text{Last}}$ | $\mathcal{A}_{\text{Avg}}$ | $\mathcal{A}_{\text{Last}}$ | $\mathcal{A}_{\text{Avg}}$ | $\mathcal{A}_{\text{Last}}$ | $\mathcal{A}_{\text{Avg}}$ | $\mathcal{A}_{\text{Last}}$ | $\mathcal{A}_{\text{Avg}}$ |
| CODA-Prompt(Smith et al., 2023) | 74.91±0.33 | 79.25±0.53 | 68.62±0.52 | 74.61±0.31 | 89.16±0.08 | 92.46±0.79 | 81.18±0.71 | 86.62±0.93 |
| SLCA(Zhang et al., 2023) | 81.01±0.11 | 84.18±0.28 | 77.23±0.40 | 82.21±0.49 | 91.30±0.54 | 93.97±0.56 | 88.54±0.35 | 92.66±0.78 |
| RanPAC(McDonnell et al., 2023) | 79.53±0.12 | 83.69±0.13 | 75.47±0.22 | 81.20±0.15 | 92.68±0.16 | 94.85±0.54 | 90.77±0.17 | 94.00±0.74 |
| OS-Prompt(Kim et al., 2024) | 75.78±0.06 | 80.49±0.49 | 72.50±0.78 | 78.43±1.07 | 88.35±0.23 | 92.04±0.76 | 81.46±0.56 | 87.15±1.15 |
| CPrompt(Gao et al., 2024b) | 77.99±0.31 | 82.34±0.32 | 73.77±0.14 | 79.81±0.50 | 89.04±0.41 | 92.26±0.57 | 84.48±0.46 | 89.35±1.09 |
| VQ-Prompt(Jiao et al., 2024) | 76.00±0.28 | 79.84±0.56 | 74.76±0.29 | 79.30±0.34 | 90.97±0.17 | 93.50±0.67 | 89.25±0.43 | 92.53±0.68 |
| EASE(Zhou et al., 2024b) | 76.75±0.41 | 81.14±0.31 | 73.90±0.61 | 80.26±0.52 | 89.53±0.15 | 92.64±0.73 | 86.30±0.36 | 90.80±0.99 |
| SSIAT(Tan et al., 2024) | 80.52±0.07 | 84.25±0.31 | 78.35±0.34 | 82.39±0.42 | 92.01±0.15 | 94.37±0.68 | 90.07±0.44 | 93.52±0.65 |
| InfLoRA+CA(Liang & Li, 2024) | 80.92±0.28 | 84.22±0.30 | 76.50±0.30 | 81.57±0.34 | 92.28±0.06 | 94.46±0.63 | 90.39±0.01 | 93.32±0.72 |
| MOS(Sun et al., 2025) | 78.76±0.17 | 82.37±0.26 | 75.16±0.59 | 80.53±0.70 | 92.31±0.20 | 94.44±0.74 | 89.43±0.37 | 92.95±0.79 |
| **Ours** | 83.37±0.26 | 87.00±0.35 | 79.43±0.34 | 84.34±0.32 | 92.40±0.08 | 94.71±0.64 | 90.51±0.14 | 93.48±0.67 |

only trains PEFT modules using data from the first task; and the fine-tuning method SLCA (Zhang et al., 2023), which also incorporates the Classifier Alignment step as in our approach.

Table 1 summarizes the performance of these SOTA methods across four widely used benchmark datasets. We report both the accuracy on the last task ($\mathcal{A}_{\text{last}}$) and the average accuracy across all tasks ($\mathcal{A}_{\text{avg}}$), presenting the mean and standard deviation over three independent runs with different random seeds. The use of random seeds introduces variability in class order across runs, making the evaluation of model performance more challenging. Notably, our method achieves superior performance in both $\mathcal{A}_{\text{last}}$ and $\mathcal{A}_{\text{avg}}$. Our method demonstrates impressive results, particularly on the more challenging datasets with larger domain gaps, such as ImageNet-R and ImageNet-A. On ImageNet-R, our method achieves a final accuracy of 81.88%, surpassing the second-

best method, SSIAT, by a significant margin of 2.33%. The $\mathcal{A}_{\text{avg}}$ also surpasses the second-best SSIAT by 2.25%. On the ImageNet-A dataset, our method achieves a final accuracy of 64.14%, surpassing the second-best SSIAT by 1.49%. These results highlight the effectiveness of our PEFT-based approach in significantly improving performance on datasets with large domain shifts, outperforming both first-session adaptation methods and full fine-tuning methods, as well as other PEFT-based approaches.

In contrast, on the CIFAR-100 and CUB-200 datasets, our method performs well, though with marginal benefits compared to other methods. Notably, on the CUB-200 dataset, our method achieves superior performance. It is also important to note that the first-session adaptation-based method, RanPAC, performs well on both CIFAR-100 and CUB-200, likely due to the significant relevance between the pretraining dataset (ImageNet) and these two datasets.

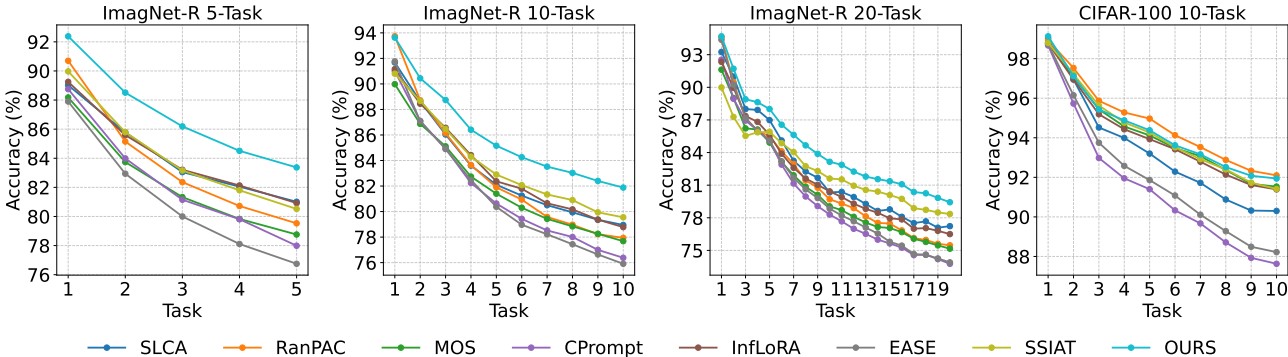

*Figure 3.* The performance of each learning session under different settings of ImageNet-R and CIFAR100. All methods are initialized with ViT-B/16-IN21k. These curves are plotted by calculating the average performance across three different seeds.

Additionally, we evaluate the performance of our method on both longer task sequences (20 tasks) and shorter task sequences (5 tasks) for CIFAR-100 and ImageNet-R, as reported in Tables 2. Across these varied experimental settings, our method consistently outperforms competing approaches, demonstrating its stability and robustness in handling diverse CIL scenarios.

Figure 3 illustrates the incremental accuracy of each session for three ImageNet-R settings and one CIFAR-100 setting. The results show that on ImageNet-R, our method consistently achieves the best performance, with a clear distinction from other methods. On CIFAR-100, our method is relatively stable, and the final results are comparable to the first-session adaptation-based method, RanPAC, which is closely related to pretraining data characteristics.

### 4.3. Ablation Study

#### 4.3.1. THE IMPACT OF EACH COMPONENT

As shown in Table 3, we systematically assess the contributions of different components to the baseline method on the ImageNet-R dataset. The baseline, consisting of a task-specific LoRA structure with angular penalty loss for classification, achieves a competitive performance of 79.36% in $\mathcal{A}_{\text{Last}}$. In exp-II, we add the MSC module, which, in conjunction with classifier alignment, provides an improvement of over 1.4%. The incorporation of Covariance Calibration with classifier alignment in exp-III also leads to an improvement of approximately 1.35%. These results underscore the importance of both mean shift compensation and covariance calibration in aligning feature distributions across tasks, thereby reducing catastrophic forgetting and enhancing stability across task sequences. Combining the MSC and CC modules gives a significant boost, improving performance by approximately 2.3% above the baseline method. Finally, the inclusion of patch distillation offers a further marginal improvement, resulting in a state-of-the-art performance of 81.88% for $\mathcal{A}_{\text{Last}}$ and 85.95% for $\mathcal{A}_{\text{Avg}}$, confirming the effectiveness of our method.

*Table 3.* The ablation studies for each component contribution evaluated on 10-session ImageNet-R. **MSC** means **M**ean **S**hift **C**ompensation. **CC** is **C**ovariance **C**alibration. **PD** indicates **P**atch **D**istillation. The **exp-I** is the **baseline** (eLoRA+$\mathcal{L}_{\text{cls}}$).

| Components⧹Ablations | MSC | CC | PD | $\mathcal{A}_{\text{Last}}$ | $\mathcal{A}_{\text{Avg}}$ |
|---|---|---|---|---|---|
| **exp-I** | ✗ | ✗ | ✗ | $79.36 \pm 0.57$ | $84.58 \pm 0.67$ |
| **exp-II** | ✔ | ✗ | ✗ | $80.81 \pm 0.16$ | $85.47 \pm 0.36$ |
| **exp-III** | ✗ | ✔ | ✗ | $80.70 \pm 0.32$ | $85.52 \pm 0.42$ |
| **exp-IV** | ✔ | ✔ | ✗ | $81.60 \pm 0.09$ | $85.88 \pm 0.35$ |
| **exp-V** | ✔ | ✔ | ✔ | $81.88 \pm 0.07$ | $85.95 \pm 0.27$ |

#### 4.3.2. LoRA STRUCTURES DESIGN

While there has been significant exploration of task-specific and task-shared PEFT modules, particularly concerning prompts and adapters, research on LoRA-based modules is relatively limited. In this paper, we investigate the use of task-specific and task-shared LoRA modules, as well as a hybrid architecture that combines both, inspired by Hydra-LoRA (Tian et al., 2024). In Table 4, we evaluate these designs across four datasets and report the final accuracy averages from three trials. The results indicate that the performance differences among the LoRA designs are minimal, with the task-specific design slightly outperforming the other two, except for the ImageNet-A dataset, where the task-shared LoRA module achieves a marginally higher performance.

*Table 4.* Experimental results of different LoRA structures. We report the final accuracy $\mathcal{A}_{\text{Last}}$. Average of three trials.

| Structure | CIFAR-100 | ImageNet-R | ImageNet-A | CUB-200 |
|---|---|---|---|---|
| G-LoRA | 91.53±0.19 | 81.15±0.19 | **64.27±0.08** | 90.16±0.56 |
| E-LoRA | **91.94±0.17** | **81.88±0.07** | 64.14±0.58 | **90.52±0.13** |
| Hydra-LoRA | 91.48±0.13 | 81.00±0.09 | 63.64±0.59 | 89.82±0.30 |

## 5. Limitations and Future Works

In this study, we have focused on addressing semantic drift by aligning first-order (mean) and second-order (covariance)

statistics. While this approach has shown promising results, it is inherently limited in its ability to capture more complex aspects of feature distribution shifts. Specifically, higher-order moments, such as skewness (third-order statistic) and kurtosis (fourth-order statistic), are not considered in this framework. These higher-order statistics could provide additional insights into the shape and tails of the data distribution, which may help in mitigating semantic drift more effectively, especially in tasks with significant feature distribution shifts. Future work will explore this approach by incorporating higher-order statistical moments like skewness and kurtosis into the alignment process.

## 6. Conclusion

We analyze catastrophic forgetting in machine learning models using parameter-efficient fine-tuning based on pre-trained models. Our experiments reveal that low-rank adaptations like LoRA induce feature mean and covariance shifts, termed Semantic Drift. To address this, we propose mean shift compensation and covariance calibration to constrain feature moments, maintaining both model stability and plasticity. Additionally, we implement feature self-distillation for patch tokens to enhance feature stability. Our task-agnostic continual learning framework outperforms existing methods across multiple public datasets.

## Acknowledgements

This work has been supported by the New Cornerstone Science Foundation through the XPLORER PRIZE, National Natural Science Foundation of China Grant No. (72188101, 62472139, U22A6001), National Key Research and Development Program of China No.2023YFE0108600, "Pioneer" and "Leading Goose" R&D Program of Zhejiang No.2024SSYS0002, the Anhui Provincial Natural Science Foundation, China (Grant No. 2408085QF191), the Fundamental Research Funds for the Central Universities (Grants No. JZ2024HGTA0178, JZ2023HGQA0097), and the Open Project Program of the State Key Laboratory of CAD&CG (Grant No. A2403), Zhejiang University.

## Impact Statement

This paper presents work whose goal is to advance the field of Machine Learning. There are many potential societal consequences of our work, none of which we feel must be specifically highlighted here.

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
