# OpenReview forum: "Navigating Semantic Drift in Task-Agnostic Class-Incremental Learning"
_ICML.cc/2025/Conference — ICML 2025 oral_

### Official Review · Reviewer_o87J · 2025-02-28

**Overall Recommendation:** 4

**Summary:**

This paper proposes a method to improve model stability in class-incremental learning by alleviating the semantic drift phenomenon. The authors leverage the transferability of pretrained models and train parameter-efficient fine-tuning LoRA modules with a frozen ViT backbone. They define semantic drift in two dimensions: the mean and covariance of the features. To address this, they compensate for shifts in the mean and introduce a novel approach that uses a Mahalanobis distance loss function to constrain the covariance. The updated first- and second-order statistics are then used to align the classifier in the post-training stage. Additionally, the authors employ a patch-token-based distillation loss to further improve the model’s performance. The method is extensively validated on four major CIL datasets, with each module demonstrating its effectiveness and achieving SOTA performance.

**Claims And Evidence:**

The authors claim that stability in class-incremental learning (CIL), which is a primary goal of CIL, can be undermined by the phenomenon of semantic drift. This phenomenon induces shifts in feature mean and covariance. The claim is supported by visualization and empirical validation across standard benchmarks.

**Essential References Not Discussed:**

No

**Ethics Expertise Needed:**

["Other expertise"]

**Experimental Designs Or Analyses:**

1. The experimental design is valid, including ablations of the effectiveness of each component, different LoRA design choices, and a comparison with a large number of very recent SOTA methods across four mainstream benchmark datasets.

2. Although the authors mention that hyperparameter choices are based on sensitivity analysis, experimental results still need to be presented.

**Methods And Evaluation Criteria:**

The authors propose using low-rank adaptation modules that continually adapt to incremental tasks, and introduce a mean shift compensation and covariance calibration method to alleviate the semantic drift phenomenon, thereby improving stability. The mean shift compensation tracks the mean shift in the feature distribution, while the covariance calibration module adds constraints to the shape of the distribution. The compensated mean and calibrated covariance matrix are then used to update the classifier head at the end of each incremental session. They also incorporate a distillation module to further improve the stability aspect of the CIL task. Overall, the method seems reasonable for addressing the CIL challenge. The proposed method is evaluated on common datasets: ImageNet-R, ImageNet-A, CUB-200, and CIFAR-100, reporting both last and average performances.

**Other Comments Or Suggestions:**

No

**Other Strengths And Weaknesses:**

Strength:
Creatively imposing the Mahalanobis distance as a constraint in the loss function, which implicitly constrains the covariance matrix.

Weakness:
The gain from patch distillation is relatively small.

**Questions For Authors:**

See experimental designs or analyses, and other strengths and weaknesses.

**Relation To Broader Scientific Literature:**

This work extends the definition of semantic drift, as described in the literature (e.g., Semantic Drift Compensation for Class-Incremental Learning), to both class mean and covariance perspectives, and proposes a novel covariance calibration method.

**Theoretical Claims:**

The mathematical symbols, variables, and equations are generally well defined and mathematically correct.

---

> ### Author Rebuttal · Authors · 2025-03-31
>
> We thank Reviewer o87J for the valuable comments. Reviewer o87J gives a **positive rating (3-Weak Accept)**, finds our method is "reasonably designed" with "a novel, creative covariance calibration approach" and mathematical formulations are "well-defined and correct", etc. We address the main concerns below:
>
> > **Reviewer o87J asks if the sensitivity analysis of the hyperparameters could be presented.**
>
> Thanks.  Here we include the sensitivity study for some important hyperparameters. The results are shown in the table below.
>
> | s    | 10         | 20         | 30         | 40         |
> | ---- | ----------| ---------- | ---------- | ---------- |
> |  $\mathcal A_{Last}$    | $79.18 \pm 0.17$ | $81.88 \pm 0.07$ | $81.27 \pm 0.07$ | $80.53 \pm 0.30$ |
>
> | r    | 8          | 16         | 32         | 64         |
> | ---- | ---------- | ---------- | ---------- | ---------- |
> |   $\mathcal A_{Last}$  | $80.80 \pm 0.27$ | $81.31 \pm 0.04$ | $81.88 \pm 0.07$ | $81.88 \pm 0.25$ |
>
> |$\lambda$| 0.2        | 0.4 | 0.6 | 0.8        |
> | ---- | ---------- | ---------- | ---------- | ---------- |
> |  $\mathcal A_{Last}$   | $81.83 \pm 0.28$ | $81.88 \pm 0.07$ | $81.72 \pm 0.18$ | $81.44 \pm 0.19$ |
>
> >**Reviewer o87J questions the effectiveness of the patch distillation module.**
>
> Thanks. Kindly refer to our response to the first concerns raised by Reviewer XH2d.

---

> > ### Comment · Reviewer_o87J · 2025-04-03
> >
> > Thank you for the valuable feedback of author. My concerns have been addressed.

---

### Official Review · Reviewer_JCMA · 2025-03-10

**Overall Recommendation:** 4

**Summary:**

This paper identifies that the feature distribution gap between novel and existing tasks is primarily influenced by differences in mean and covariance moments. To address this, a novel semantic drift calibration method is proposed, integrating mean shift compensation and covariance alignment. Specifically, a Mahalanobis distance constraint is applied to align class-specific embedding covariances between previous and current networks, effectively mitigating covariance shift. Additionally, a feature-level self-distillation mechanism is introduced to enhance generalization and improve model adaptation.

**Claims And Evidence:**

The claims in the submission are well-supported by compelling empirical evidence. The proposed method exhibits superior performance, particularly on challenging benchmarks like ImageNet-R and ImageNet-A, where it outperforms the second-best method, SSIAT, by substantial margins in both 𝐴_{last} and 𝐴_{avg}. Ablation studies validate the contributions of key components, Mean Shift Compensation (MSC) and Covariance Calibration (CC), which collectively improve performance remarkably. Furthermore, the method demonstrates robustness across both long and short task sequences, reinforcing its reliability in diverse class-incremental learning (CIL) scenarios.

**Essential References Not Discussed:**

No

**Experimental Designs Or Analyses:**

The experimental design is generally well-structured and appropriate for evaluating class-incremental learning. The use of challenging datasets, ablation studies, and comparisons with a strong baseline strengthen the validity of the findings.

**Methods And Evaluation Criteria:**

The proposed methods and evaluation criteria appear well-aligned with the problem of class-incremental learning (CIL). The use of challenging benchmarks such as ImageNet-R and ImageNet-A is appropriate, as these datasets introduce real-world complexities, including distribution shifts and adversarial robustness, which test the model’s adaptability. The evaluation metrics, 𝐴_{last} and 𝐴_{avg}, effectively capture both final and average performance across task sequences, providing a comprehensive assessment of catastrophic forgetting and knowledge retention. Additionally, the inclusion of ablation studies on Mean Shift Compensation (MSC) and Covariance Calibration (CC) further validates the method’s contribution. The experiments on both long and short task sequences enhance the evaluation’s credibility by ensuring the method’s robustness across varying levels of incremental complexity. Overall, the methods and evaluation criteria are well-designed for the intended application, effectively assessing both adaptability and stability in CIL scenarios.

**Other Comments Or Suggestions:**

None

**Other Strengths And Weaknesses:**

None

**Questions For Authors:**

1. The results in Table 2 show that the performance on ImageNet-R is good but seems not good enough on CIFAR-100. Could you provide a detailed explanation for this discrepancy?

2. Better to discuss the limitations and future studies.

**Relation To Broader Scientific Literature:**

This paper contributes to class-incremental learning (CIL) by integrating LoRA-based fine-tuning with semantic drift compensation, covariance calibration, classifier alignment, and feature self-distillation, addressing key challenges in parameter-efficient adaptation. It builds upon prior research in parameter-efficient fine-tuning (PEFT) for CIL\cite{hu2021lora, valipour2022dylora, hao2024flora}, which focuses on reducing computational overhead while preserving model adaptability. Unlike standard PEFT methods, this work explicitly tackles feature drift, aligning with studies on feature shift correction\cite{liu2021swin, zhu2024vision} by introducing covariance alignment and Mahalanobis distance constraints to stabilize class representations over incremental tasks. Additionally, the paper extends classifier bias mitigation\cite{wu2019large, kang2019decoupling} by ensuring that learned features remain well-calibrated across tasks without reliance on memory buffers. By integrating feature-level self-distillation, it also aligns with research in self-supervised learning and knowledge retention\cite{zhang2022self, touvron2021training}. These innovations improve both efficiency and stability in incremental learning, making the proposed approach highly relevant to the broader literature on continual learning, parameter-efficient adaptation, and large-scale vision model optimization.

**Theoretical Claims:**

The submission does not include formal theoretical proofs but presents technically grounded mathematical formulations. The formulations for Mean Shift Compensation (MSC) and Covariance Calibration (CC) are well-justified, particularly in their use of Mahalanobis distance constraints for covariance alignment. Additionally, the loss functions and optimization strategy align with established techniques for mitigating catastrophic forgetting. While no theoretical analysis is provided, the effectiveness of these formulations is validated through empirical experiments and ablation studies, demonstrating their practical impact. Overall, the mathematical foundations of the proposed method are sound and effectively supported by experimental results.

---

> ### Author Rebuttal · Authors · 2025-03-31
>
> We thank Reviewer JCMA for the valuable comments. Reviewer JCMA gives a **positive rating (4-Accept)**, acknowledges "a novel semantic drift calibration method" which is "well-designed and effectively assessed" and notes that its "key components improve performance remarkably" offering "improved efficiency and stability", etc. We address the main concerns below:
>
> > **Reviewer JCMA asks if we could provide a detailed explanation for the performance discrepancy in CIFAR-100 and Imagenet-R.**
>
> Thanks. Please refer to our response to the second concerns raised by Reviewer XH2d.
>
> > **Reviewer JCMA asks if we could discuss the limitations and future studies.**
>
> Thanks. We discuss the Limitations and Future Work below. In this study, we have focused on addressing semantic drift by aligning first-order (mean) and second-order (covariance) statistics. While this approach has shown promising results, it is inherently limited in its ability to capture more complex aspects of feature distribution shifts. Specifically, higher-order moments, such as skewness (third-order statistic) and kurtosis (fourth-order statistic), are not considered in this framework. These higher-order statistics could provide additional insights into the shape and tails of the data distribution, which may help in mitigating semantic drift more effectively, especially in tasks with significant feature distribution shifts. Future work will explore this approach by incorporating higher-order statistical moments like skewness and kurtosis into the alignment process.

---

### Official Review · Reviewer_XH2d · 2025-03-13

**Overall Recommendation:** 4

**Summary:**

Balancing flexibility and stability remain a key challenge in class-incremental learning (CIL). To address this, this paper introduces mean shift compensation and covariance calibration to regulate feature moments, preserving both model stability and adaptability. Additionally, a feature self-distillation mechanism for patch tokens is implemented to further enhance feature consistency. The proposed efficient, task-agnostic continual learning framework surpasses existing methods across multiple public datasets, demonstrating its effectiveness and superiority.

**Claims And Evidence:**

The claims made in the submission are well-supported by clear and convincing evidence. The proposed mean shift compensation, covariance calibration, and feature self-distillation methods are validated through extensive experiments across multiple public datasets. The results consistently demonstrate superior performance over existing class-incremental learning (CIL) methods.

**Essential References Not Discussed:**

None.

**Experimental Designs Or Analyses:**

The experimental design is well-structured and comprehensive, covering component-wise ablations, model design choices, and competitive benchmarking.

**Methods And Evaluation Criteria:**

The proposed methods and evaluation criteria are well-suited for the class-incremental learning (CIL) problem. address semantic drift through mean shift compensation and covariance calibration, ensuring both stability and adaptability. The mean shift compensation tracks distribution shifts, while covariance calibration constrains feature variance, collectively enhancing classifier alignment across incremental tasks. Additionally, a feature self-distillation module further stabilizes feature representations over time.

**Other Comments Or Suggestions:**

None.

**Other Strengths And Weaknesses:**

Strength:
1. Good writing with well-organized structure.
2. The illustration looks good and intuitive.
3. The performance looks good compared with SOTA approaches.
4. The motivation is clear and straightforward and well demonstrated with carefully designed experiments .

Weakness:
The patch distillation module appears to be somewhat incremental and lacks a strong integration with the overall motivation and methodology. While it aims to enhance feature stability, its connection to the core semantic drift calibration and covariance alignment framework is not entirely clear. Specifically, its role in mitigating catastrophic forgetting or improving class-incremental learning (CIL) performance should be more explicitly justified. If it primarily contributes to feature refinement rather than directly addressing semantic drift or classifier calibration, further clarification on its necessity and integration with the main approach is needed.

**Questions For Authors:**

The gains on CIFAR-100 seem marginal. Could you please provide a detailed discussion on this?

**Relation To Broader Scientific Literature:**

This work mitigates the Semantic Drift issues in CIL by integrating both mean shift compensation and covariance calibration. It builds upon prior research in parameter-efficient fine-tuning (PEFT) for CIL~\cite{hu2021lora, valipour2022dylora, hao2024flora}, which focuses on reducing computational overhead while preserving model adaptability. Existing works only study the semantic drift phenomenon by exploring class prototypes shifting, however this work extends the semantic drift phenomenon in both the mean and covariance, calibrating them to mitigate catastrophic forgetting.

**Theoretical Claims:**

The formulations are mathematically technically correct.

---

> ### Author Rebuttal · Authors · 2025-03-31
>
> We thank Reviewer XH2d for the valuable comments. Reviewer XH2d gives a **positive rating (4-Accept)**, finds that "the paper is well-written and well-organized" with "a clear, straightforward motivation" coupled with "intuitive illustrations" and "demonstrate superior performance" with "careful experimental designs", etc. We address the main concerns below:
>
> > **Reviewer XH2d asks for further clarification on the necessity of integrating the patch distillation module with the main approach that alleviates semantic drift.**
>
> Thanks. Patch distillation is not an independent add-on separate from semantic drift calibration; rather, it enhances feature space stability to provide a more reliable basis for mean/covariance calibration. Its mechanism is as follows: semantic drift calibration (MSC/CC) targets high-level semantic features (class tokens), while patch tokens capture fine-grained visual details. Stabilizing local feature representations can reduce the distributional shift at the class token level. In ViT, shallow features encompass general patterns (such as edges and textures) shared across tasks. Enforcing consistency among patch tokens helps prevent the degradation of these foundational features. In summary, Patch Distillation is not an isolated or unrelated component; rather, it complements MSC/CC and plays a crucial role in improving model robustness.
>
> > **Reviewer XH2d asks for a detailed discussion on the gains of our method on CIFAR-100.**
>
> Thanks. We highlight that our approach is designed to be general. Notably, it demonstrates significant advantages on more challenging tasks, such as ImageNet-R and ImageNet-A. In our experiments, we applied a unified hyper-parameter setting without dataset-specific tuning, which underscores the versatility of our approach. We are confident that by fine-tuning the hyperparameters of our method for CIFAR-100, further improvements can be achieved.  Within a limited timeframe, we find an optimized hyperparameter configuration using LoRA with a rank of  $r = 20$ and a patch distillation loss weight of  $\lambda = 0.1$. This setting achieve a CIFAR-100 performance of $\mathcal A_{\text{Last}} = 92.20$ and $\mathcal A_{\text{Avg}} = 95.15$.

---

### Official Review · Reviewer_31F2 · 2025-03-13

**Overall Recommendation:** 4

**Summary:**

This paper tackles the challenge of class-incremental learning in continual Learning, which enables models to sequentially learn multiple tasks without retraining or accessing data from previous tasks.
While recent advancements in deep learning, such as larger model capacities and large-scale pretraining, have improved plasticity, traditional methods (e.g., regularization, memory replay, and knowledge distillation) still come with significant computational and storage overheads, hindering practical deployment.
The authors focus on addressing catastrophic forgetting and the issue of semantic drift, which occurs when feature means and covariances shift as new tasks are added.

They propose two solutions:
(1) Mean Shift Compensation, which estimates and corrects the drift in feature means by computing weighted averages of embedding shifts, and (2) Covariance Calibration, which uses Mahalanobis distance to align the covariance matrices of embeddings from old and new tasks. These methods are incorporated into a task-agnostic continual learning framework that outperforms existing techniques across several public datasets, enhancing both model stability and adaptability.

**Claims And Evidence:**

The claims made in the submission are generally well-supported by clear and convincing evidence. The proposed method demonstrates superior performance, especially on challenging datasets like ImageNet-R and ImageNet-A, where it surpasses the second-best method, SSIAT, by significant margins in both A_{last} and A_{avg}. Additionally, the ablation study confirms the effectiveness of key components like Mean Shift Compensation (MSC) and Covariance Calibration (CC), which together improve performance by over 2%. The method’s robustness across task sequences (both long and short) further solidifies its reliability in diverse class-incremental learning (CIL) scenarios.

**Essential References Not Discussed:**

No

**Experimental Designs Or Analyses:**

The experimental design is well-structured and rigorous, employing diverse and challenging benchmark datasets, consistent task sequences, and fair comparisons with state-of-the-art methods, ensuring a comprehensive evaluation of model performance. The study’s use of multiple independent runs with identical random seeds enhances result reliability, while the ablation studies effectively isolate the contributions of key components, particularly Mean Shift Compensation (MSC) and Covariance Calibration (CC), demonstrating their effectiveness in mitigating semantic drift. The inclusion of both short (5 tasks) and long (20 tasks) sequences further validates the method’s robustness across different continual learning settings. Additionally, the framework’s reliance on pretrained models with LoRA-based fine-tuning presents a computationally efficient alternative to full fine-tuning approaches. While statistical significance testing and computational cost analysis could further reinforce the claims, the experimental design provides strong empirical evidence supporting the proposed method’s superiority in handling catastrophic forgetting and domain shifts, making it a promising solution for class-incremental learning.

**Methods And Evaluation Criteria:**

The proposed methods and evaluation criteria make sense for the problem of class-incremental learning (CIL) and align well with the challenges inherent in this application. The approach leverages a frozen ViT backbone with task-specific LoRA modules, which is an effective way to retain previous knowledge while allowing task adaptation without excessive memory overhead. LoRA’s ability to integrate low-rank weight updates ensures that task-specific adaptation is computationally efficient, making it well-suited for CIL. Semantic drift (feature mean and covariance shift) is a known challenge in CIL, and the method addresses it explicitly through: Mean Shift Compensation (MSC): Correcting shifts in class mean embeddings. Covariance Calibration (CC): Aligning covariance matrices using Mahalanobis distance, ensuring feature distributions remain consistent across tasks. These techniques align with prior findings on distributional shifts in continual learning and provide a principled way to improve knowledge retention. Results highlight the method’s strengths in handling domain shifts (ImageNet-R, ImageNet-A) while performing competitively on natural datasets (CIFAR-100, CUB-200).

**Other Comments Or Suggestions:**

No

**Other Strengths And Weaknesses:**

No

**Questions For Authors:**

1. How does the proposed LoRA-based fine-tuning scale across different model sizes with varying complexities? In particular, how does it handle imbalanced class distributions?

**Relation To Broader Scientific Literature:**

The paper advances the field by integrating LoRA-based fine-tuning with semantic drift compensation, covariance calibration, classifier alignment, and feature self-distillation, all of which address major challenges in CIL. It builds on prior work in PEFT-based CIL, feature shift correction, classifier bias mitigation, and ViT self-distillation, but improves upon them by removing reliance on memory buffers, reducing computational overhead, and enhancing feature stability. These contributions align with and extend existing literature, making them highly relevant to the broader research community in continual learning and parameter-efficient adaptation of large models.

**Theoretical Claims:**

Overall, the theoretical claims appear reasonable and grounded.

---

> ### Author Rebuttal · Authors · 2025-03-31
>
> We thank Reviewer 31F2 for the valuable comments. Reviewer 31F2 gives a **positive rating (3-Weak Accept)**, finds "the proposed method demonstrates superior performance", and our experimental designs are "rigorous, diverse and comprehensive", etc. We address the main concerns below:
>
> > **Reviewer 31F2 suggests that a computational cost analysis would be useful to demonstrate that LoRA-based tuning is more efficient in terms of computation than full fine-tuning.**
>
> Thanks. We evaluate the computational cost in terms of the number of trainable parameters and Multiply-Accumulates (MACs). We analyze the computational cost of multiple SOTA methods on ImageNet-R. The results are shown in the table below.
>
> |Method| Trainable Params (M) | MACs(G) |ImageNet-R ($\mathcal A_{Last}$)|
> |---|---|---|---|
> |L2P|0.19|37.48|70.56|
> |DualPrompt|0.41|35.17|66.89|
> |RanPAC|2.00|17.58|77.94|
> |SLCA|85.8|17.58|78.95|
> |CPrompt|0.26|25.01|76.38|
> |EASE|1.19|17.81|75.91|
> |MOS|3.2|17.64|77.68|
> |InfLoRA|0.32|20.51|78.78|
> |SSIAT|1.19|17.81|79.55|
> |Ours|1.20|20.82|81.88|
>
> The results highlight that our Params and MACs are minimal. Even under low computational cost conditions across all methods, our approach still delivers outstanding results.
>
> > **Reviewer 31F2 asks how our method scales across different model sizes with varying complexities.**
>
> Thanks. We evaluate our method with four different scales of models pretrained on ImageNet-21K at a resolution of $224 \times 224$.
> |Model|depth|embed_dim|heads|Params(M)| $$\mathcal A_{Last}$$ | $$\mathcal A_{Avg}$$ |
> |---|---|---|---|---|---------- |---------- |
> | ViT-Ti/16 |12|192|3|5.7|$60.85$|$70.90$|
> | ViT-S/16 |12|384|6|22|$75.82$|$81.87$|
> | ViT-B/16 |12|768|12|86| $81.83$|$86.27$ |
> | ViT-L/16 |24|1024|16|307| $85.28$ | $89.10$ |
>
> The results show that our method demonstrates strong scalability with pre-trained model size. The  ViT-Large based model can get better performance.
>
> > **Reviewer 31F2 asks how our method handles imbalanced class distributions.**
>
> Thanks. We identify a notable class imbalance in our training datasets, such as ImageNet-R and ImageNet-A. For ImageNet-R, the training set comprises 200 classes with a total of 24,000 samples. The most frequent class contains 349 samples, whereas the least frequent class contains only 38 samples, resulting in a maximum-to-minimum sample ratio of approximately 9.18:1. In comparison, the ImageNet-A training set also consists of 200 classes but includes only 5,981 samples. Here, the most common class has 86 samples, while the least common class has just 2 samples, leading to a ratio of 43:1. Our proposed method is capable of addressing this class imbalance issue during training.
>
> Specifically, in the classifier alignment phase, we employ the following strategy to alleviate the class imbalance:
> For each class $ c $, we generate $s_c$ synthetic feature samples from a normal distribution $ \mathcal N(\mu_c, \Sigma_c) $, where $s_c$ is the same for all classes, and thus independent of class frequency. This design ensures that minority classes generate more synthetic samples relative to their original size, effectively mitigating the classifier’s bias toward high-frequency classes.
>
> Class imbalance itself is an important research direction, and we consider extending our method to better address this challenge as a promising avenue for future work.

---

> > ### Comment · Reviewer_31F2 · 2025-04-03
> >
> > Thank authors for the additional experiments and explanations. The impressive performance on the ImageNet dataset makes the work appear solid. I have also carefully revisited the remaining aspects and will accordingly raise my score further.

---

### Decision · Program_Chairs · 2025-05-01

**Decision:**

Accept (oral)

**Comment:**

This paper introduces a novel solution to semantic drift in class-incremental learning (CIL) by addressing shifts in feature mean and covariance between new and old tasks. The authors propose Mean Shift Compensation (MSC) and Covariance Calibration (CC), using Mahalanobis distance constraints to align embedding covariances and reduce drift. Additionally, a patch-token-based self-distillation module is incorporated to enhance model stability, and the method is validated on several benchmark datasets, showing improved performance.

The strengths of the paper lie in its innovative solution to semantic drift through the combination of MSC and CC, demonstrating both theoretical soundness and strong empirical results. The approach performs particularly well on challenging datasets like ImageNet-R and ImageNet-A and shows the effectiveness of its key components through ablation studies.